# Comparing and integrating human mobility data sources for measles transmission modeling in Zambia

Natalya Kostandova[1]*, Christine Prosperi[2], Simon Mutembo[2], Chola Nakazwe[3], Harriet Namukoko[4], Bertha Nachinga[3], Shengjie Lai[5], Andrew J Tatem[5], Qianwen Duan[5], Elliot N Kabalo[6], Kabondo Makungo[7], Gershom Chongwe[8], Innocent Chilumba[9], Gloria Musukwa[10], Kalumbu H Matakala[11], Mutinta Hamahuwa[11], Webster Mufwambi[12], Japhet Matoba[13], Irene Mutale[9], Kenny Situtu[9], Edgar Simulundu[10], Phillimon Ndubani[10], Alvira Z Hasan[2], Shaun A Truelove[1,2], Amy K Winter[14,15], Andrea C Carcelen[2], Bryan Lau[1], William J Moss[1,2], Amy Wesolowski[1]

1 Department of Epidemiology, Johns Hopkins Bloomberg School of Public Health, Baltimore, Maryland, United States of America, 2 Department of International Health, International Vaccine Access Center, Johns Hopkins Bloomberg School of Public Health, Baltimore, Maryland, United States of America, 3 Information, Research and Dissemination, Zambia Statistics Agency, Lusaka, Zambia, 4 Population and Social Statistics, Zambia Statistics Agency, Lusaka, Zambia, 5 WorldPop, School of Geography and Environmental Science, University of Southampton, Southampton, United Kingdom, 6 Zambia Information and Communications Technology Authority, Lusaka, Zambia, 7 Zamtel, Lusaka, Zambia, 8 Tropical Diseases Research Centre, Ndola, Zambia, 9 Biomedical Sciences Department, Tropical Diseases Research Centre, Ndola, Zambia, 10 Administration, Macha Research Trust, Choma, Zambia, 11 Clinical Research Department, Macha Research Trust, Choma, Zambia, 12 Administration, Tropical Diseases Research Centre, Ndola, Zambia, 13 Molecular Biology Department, Macha Research Trust, Choma, Zambia, 14 Department of Epidemiology and Biostatistics, University of Georgia, Athens, Georgia, United States of America, 15 Center for the Ecology of Infectious Diseases, University of Georgia, Athens, Georgia, United States of America

* nkostan1@jh.edu

## Abstract

Quantifying population mobility is crucial in developing accurate models of infectious disease dynamics. Increasingly, multiple data sources are available to describe individual and population mobility in a single location; however, there are no methods to systematically integrate these data. Combining information from these data sets may be valuable and help mitigate inherent biases in each data set due to sampling, censoring, and recall. We examined four commonly used data sources (mobile phone records, travel survey, Demographic and Health Survey, and Facebook location information) to quantify subnational travel patterns in Zambia. First, we explored summary metrics of mobility from each data set. Estimates of the probability of a trip and location of travel varied across data sets, with some data quantifying twice the frequency of travel as others. Then, we developed a departure-diffusion model that is able to produce a single estimate of travel by combining these data sets. When multi-data set models included mobile phone records, this data source dominated estimates given the broader spatial coverage. We then used a metapopulation model to simulate a measles outbreak to identify how these different data sets and models would

**Data availability statement:** All code, as well as mobility matrices derived from all sources of mobility data are made available at https://github.com/nkostandova/comparability_mobility_data. In addition, 2018 Zambia DHS data are publicly available (https://dhsprogram.com/methodology/survey/survey-display-542.cfm).

**Funding:** National Institute of Allergy and Infectious Diseases (1R01AI160780-01) (AW, NK, SL, AJT, ENK, KM); National Institutes of Health (DP2LM013102) (AW, NK); Bill and Melinda Gates Foundation (OPP1094816) (NK, CP, SM, CND, HN, BN, GC, IC, KHM, GM, MH, WM, JM, IM, ES, PN, AZH, SAT, AKW, ACC, WJM); National Institute of Health Ruth Kirschstein NRSA (T32) award (NK); and Burroughs Wellcome Fund (1015823.03) (AW). Funders had no role in study design, analysis, or decision to publish.

**Competing interests:** The authors have declared that no competing interests exist.

impact estimates of the spatial spread of a highly infectious disease. We found that using travel survey data to parameterize mobility resulted in the introduction of cases in 98% of districts compared to less than 50% when mobile phone data or Facebook data were used. This study highlights the need for methods that facilitate integrating multiple data sets to improve validity of mobility estimates and resultant infectious disease transmission dynamics.

## Introduction

Population mobility drives the spread of infectious diseases and plays an important role in introducing pathogens into susceptible populations and maintaining transmission for a wide range of pathogens [1,2], including measles [3–5], cholera [6,7], malaria [8–10], trachoma [11], human immunodeficiency virus [12,13], rubella [14], and influenza [15,16], among others. Use of different sources of data to quantify mobility has particularly increased since the beginning of the SARS-CoV-2 pandemic [17], with population mobility used to inform questions across different phases of public health response, including prior to an outbreak, during emergence, during outbreak, and elimination [18]. Multiple data sources have been used to quantify human mobility, including, but not limited to, surveys, mobile phone data (e.g., call detail records), Global Positioning System (GPS) loggers and other Global Navigation Satellite Systems, transportation flows (e.g., airline, railroad, sea, and subway), national censuses, and online social network data [2,19–22]. Often, a single mobility data source has been used despite spatial, temporal, and population differences across these data [23]. For instance, national population-based household surveys like the Demographic and Health Surveys (DHS) are designed to be representative of the population of interest but are collected at infrequent temporal intervals. National population and housing censuses are intended to capture mobility data from all individuals within a country, but also at infrequent (often decennial) intervals. In contrast, call data records from mobile phones only measure the movement of mobile phone subscribers, which can lead to sparse coverage, especially in rural areas, but can do so on fine temporal and spatial scales. Data sets also vary by the type of travel quantified. For example, travel surveys traditionally rely on participants' recall of trips taken, often over a long period, whereas mobile phone data have been used to quantify shifts in mobile phone tower usage by subscribers that serve as a proxy for travel. Overall, variation in these characteristics across data sets may result in different estimates of mobility due to differences in the sampled population, measurement method, and quantified type of travel.

Increasingly, diverse mobility data sources have become available in the exact geographical location. This provides an opportunity to describe mobility and, importantly, identify how different data sets may impact estimates of spatial patterns of disease transmission. In addition, this may enable data sets to be adjusted to account for biases or be otherwise combined to obtain an integrated data set with more accurate or higher spatial or temporal coverage. While the availability of multiple

data sources provides an opportunity to leverage unique information provided independently [24], challenges persist in accounting for disparate spatial and temporal scales and characteristics of individuals captured within each data set.

In this study, we aimed to assess 1) the comparability between estimates of human mobility across districts in Zambia obtained from four data sources; 2) the feasibility of using pooling and a Bayesian framework to combine information from multiple data sets; and 3) the extent to which the choice of mobility data set or combination of data sets affected estimates of introduction events using a measles virus transmission simulation in Zambia as an example. Measles was selected as an illustrative example given Zambia's remarkable progress towards elimination since the introduction of nationwide supplementary immunization activities in 2003 [25], and potential importance of identifying areas at high risk for introduction events to better target national vaccination strategy, including subnational campaigns. To achieve these objectives, we first compared probabilities of departure and diffusion from the 2018 Zambia DHS, call detail records from a major mobile phone network provider in Zambia, Facebook mobile application data, and an individual-level travel survey implemented in two districts of the country. We then used pooling and a Bayesian framework to combine information across data sets. In the Bayesian framework, probabilities of departure and diffusion from one data set were used as a prior when fitting probabilities of departure and diffusion from another data set to obtain a posterior that was informed by both data sets. Finally, we used a metapopulation compartmental model to simulate dynamics of measles virus across Zambia, and compared the estimates of introduction events occurring when different mobility data sets were used to inform population movement.

## Materials and Methods

### Mobility data sets

We analyzed four data sets describing human mobility patterns across Zambia (see S1 Table, S1 Fig).

**Travel survey.** A travel survey was nested in a serological survey of IgG antibodies to measles virus in two districts in Zambia (Choma and Ndola) from March – June 2022 [26]. In total, 3577 individuals across 72 clusters and 1245 households were surveyed. The survey collected information on socio-demographic characteristics of participants, mobile phone ownership, and overnight travel in the prior two months, including the number of trips taken, district(s) traveled to, the reason for travel, and whether the respondent was accompanied by adults or children. To obtain the number of monthly trips taken between districts of origin (Choma or Ndola) and destination districts, the reported number of trips between each pair of origins and destinations taken in the last two months was divided by two.

**Demographic and Health Survey (DHS).** The most recent Zambia DHS was carried out in 2018 and was implemented using a stratified two-stage sample design [27]. See [27] for a full description of the DHS methodology and questionnaires. In this analysis, the relevant questionnaire modules included socio-demographic characteristics of participants, mobile phone ownership, internet use within the last 12 months, and overnight travel in the last 12 months. Using the jittered GPS coordinates per household, each respondent household was matched to the corresponding district. A sampling-weighted estimate of overnight travel in the last 12 months was obtained for each district and divided by 12 to produce the average monthly probability of at least one overnight trip.

**Mobile phone data.** A spatially and temporally aggregated data set of travel between districts was obtained from the Zambia Telecommunications (Zamtel) Company Limited, one of the primary phone operators in Zambia. Based on the number of daily subscribers provided by Zamtel during this period, the highest number of subscribers on a single day corresponds to 10.5% of 2020 population of Zambia, ranging from 0% to over 40% in some districts. From March 1, 2020, to December 30, 2020, the number of Zamtel anonymized SIM IDs for which the primary location changed from one district to another between subsequent days was recorded. SIM IDs with primary cell tower locations in the same district for subsequent days were considered to have not left the district ("stays"). Cell towers were aggregated to the district level using the 2018 administrative boundaries. Due to temporal fluctuations in travel during the study period, including changes in restrictions during the COVID-19 pandemic, we further aggregated mobile phone data to obtain the total number of stays and the total number of trips between districts over the nine-month period.

**Facebook data at Meta.** The mobility data of Facebook users in Zambia were obtained from the Data for Good (DfG) Programme at Meta during the COVID-19 pandemic from April 1, 2020, through May 21, 2022. Meta provided aggregated, de-identified demographic data for supporting the responses to epidemics (e.g., COVID-19 pandemic) and natural disasters (e.g., earthquakes and floodings). These data were limited to mobile devices with the Facebook application installed that had opted to share geolocation. Details of this data set are available elsewhere [28,29]. Data were aggregated spatially to Bing tile level 12 by Meta, approximately 9.8 km at the equator [30]. For each 8-hour period, the number of individual devices that moved from one tile to another across adjacent 8-hour periods was collected. The user's location was assigned to the tile in which they appeared most frequently during each 8-hour window, or to the last tile if the frequency was the same. In the tile-level data set made available for this study, the number of individuals moving from one tile to another was discarded if fewer than ten trips were observed. Data were then aggregated to the district level, so individuals who remained in the same district across adjacent 8-hour periods were considered a "stay". To account for potential fluctuations in travel due to the COVID-19 pandemic, we aggregated the Facebook mobility data to obtain the total number of stays and the total number of trips taken between districts over the period reported.

We only had access to the anonymized, aggregated data from the DfG Programme at Meta, and all direct identifiers, as well as any characteristics that might lead to identification, were omitted from the data before we obtained them. Our analysis method complied with the DfG Data License Agreement signed, and Ethical clearance for collecting and analyzing mobility data in this study was also granted by the institutional review boards (See ethics statement section).

## Weighting mobile phone and Facebook data to account for sampling bias

Using the DHS data, for each district, we calculated a sampling-weighted probability of departure stratified by phone ownership or internet use, as well as the proportion of individuals who owned a mobile phone and/or reported using the internet. We then used these estimates to weigh the mobile phone and Facebook mobility data sets to better capture travel by individuals who did not have a mobile phone and did not use the internet in the last 12 months, respectively. For example, we used the following approach to calculate the weighted mobile phone probability of departure:

$$P(t=1)_{wMP} = P(t=1)_{MP}P(mp=1)_{DHS} + P(t=1)_{MP}\frac{P(t=1|mp=0)_{DHS}}{P(t=1|mp=1)_{DHS}}P(mp=0)_{DHS}$$

Where $P(t=1)$ is the probability of departure; $P(mp=1)$ is the probability of owning a mobile phone; $P(mp=0)$ is the probability of not owning a mobile phone; and $P(t=1|mp=i)$ is the probability of departure conditional on mobile phone ownership. Subscripts indicate the data set, where $MP$ is mobile phone data, and $wMP$ is weighted mobile phone data. Note that we only collected mobile phone data from a single operator but are taking the simplifying assumption that the ratio $\frac{P(t=1|mp=0)_{DHS}}{P(t=1|mp=1)_{DHS}}$ captures the relationship between the travel of Zamtel users and non-Zamtel users. The same approach was used to weight the Facebook data using Internet use rather than mobile phone ownership.

We recognize that this is likely to only partially account for sampling bias, especially as the DHS does not capture use of Facebook app on smartphone, which is a requirement for being included in the Facebook mobility data.

## Fitting the probability of departure

For each district, we calculated the probability of departure (i.e., a trip outside of the origin district) for each data set using the monthly average or estimated value. For each of the four data sets, we independently fit a hierarchical beta-binomial model using the *mobility* package in R [31]. Namely, for district $i$, the probability of departure is defined as

$$\tau_i \sim Beta(1+\alpha,\ 1+\beta)$$

With $\alpha$ and $\beta$ drawn from Gamma distributions:

$$\alpha \sim Gamma(0.01, \ 0.01)$$

$$\beta \sim Gamma(0.01, \ 0.01)$$

For districts that were not covered in the data set, the model assigns a population-level probability of departure, with population mean $\tau_p$ drawn from a Beta distribution with population-level shape and rate parameters $\overline{\alpha}$ and $\overline{\beta}$:

$$\tau_p \sim Beta\left(1 + \overline{\alpha}, \ 1 + \overline{\beta}\right)$$

Models were fit using the Markov Chain Monte Carlo algorithm using *mobility* package in R [31]. More details about the model and fitting procedure are available elsewhere [32].

## Obtaining pooled estimates for the probability of departure

We pooled fitted estimates for the probability of departure from all four data sets in either the raw (DHS, travel survey, mobile phone, Facebook) or weighted (DHS, travel survey, weighted mobile phone, weighted Facebook) forms. We used a random-effect model, as different data sources capture information from different populations due to disparate time and geographic coverage, as well as sampling strategies [33]. Pooling was done with the inverse variance method using the *meta* package in R [34]. For each district, we obtained a point estimate and standard error for the pooled probability of departure.

## Modeling diffusion patterns

We analyzed how the various data sets performed in estimating the destination district for a trip, given that an individual departed their district of origin. This analysis excluded the DHS data since it did not collect information on the travel destination. We first fit gravity models with different structures and selected a best-fit model for each data set, assessing model fit using goodness of fit metrics (deviance information criterion, root-mean-square error, mean absolute percentage error, and $R^2$) and summary plots for the fitted models. Key summaries of metrics for gravity models are presented in S1 Text. For the best-fitting model selected, we compared coefficient estimates across the three data sets.

To incorporate information on mobility from more than one data set, we used a Bayesian framework. While comprehensive description of Bayesian statistics is available elsewhere [35], in short, this framework relies on Bayes' rule to incorporate information both based on the observed data (the sample) and previous information (e.g., distribution of parameters obtained from previous studies, or the prior) to obtain updated distribution for the parameters of interest (the posterior). A presentation of the Bayes' rule as well as an illustrative figure demonstrating an example where information about prior distribution has an observable effect on the posterior distribution are provided in S2 Text.

We used estimated coefficients and posterior distributions from the model fit to one data set as a prior for fitting the model to another data set. We modified the source code of the *mobility* package to include an informative prior, where population and distance parameter priors for travel between two districts were drawn from a truncated normal distribution with mean and precision set to estimates of coefficients from the diffusion model fit to another data set.

In other words, for an exponential gravity model, which was the best fit model of dispersion, the number of trips from district $i$ to district $j$, $T_{ij}$, was defined by

$$T_{ij} \sim Poisson(\mu_{ij})$$

$$\mu_{ij} = \frac{\theta N_i^{\omega_i} N_j^{\omega_j}}{e^{d_{ij}/\delta}},$$

where $T_{ij}$ is the number of trips from district $i$ to district $j$, with $i$ and $j$ distinct; $N_i$ is the population size of the origin district; $N_j$ is the population size of the destination district; and $d_{ij}$ is the distance between the centroids of the two districts, in kilometers. The $\omega_i$ and $\omega_j$ parameters modify the contribution of origin and destination population sizes, respectively, while $\delta$ is the distance deterrent parameter, modifying the extent to which distance between origin and destination reduces the probability of travel, while $\theta$ is the scaling parameter. To combine information from two data sets, we used information on the distribution of fitted parameters from the first data set to generate the following priors:

$$\omega_1 \sim \ Truncnorm(mean(\hat{\omega}_1), \ sd(\hat{\omega}_1))$$

$$\omega_2 \sim \ Truncnorm(mean(\hat{\omega}_2), \ sd(\hat{\omega}_2)$$

$$\delta \sim \ Truncnorm(mean\left(\hat{\delta}\right), \ sd\left(\hat{\delta}\right))$$

Where $mean\left(\hat{\phi}\right)$ and $sd\left(\hat{\phi}\right)$ are the mean and standard deviation of the distribution for the fitted $\phi$ parameter obtained using data from the first source.

Then, data from the second source was used as sample, and a posterior was obtained for each of the parameters.

Models were implemented using *rjags* library in R [36].

To evaluate how precise coefficient estimates from a model fit to one data set must be to be an informative prior, we reduced the standard deviation on the coefficients by a single factor. We then drew values of coefficient from distributions with fitted mean and the scaled standard deviation and used those as a prior in the Bayesian approach. The second data set used in the Bayesian framework as the sample remained unchanged.

## Evaluating the impact of various mobility matrices informed by each data set on measles virus transmission

We constructed a discrete-time spatial stochastic model of measles virus transmission. We focused on measles as a case study due to the well-understood transmission dynamics, low levels of community transmission in Zambia prior to 2020, and interest of the Zambia Ministry of Health in identifying subnational areas for targeted interventions like Supplementary Immunization Activities.

Briefly, in our model, individuals transitioned between Maternal Immunity, Susceptible, Infected, Vaccinated, and Removed compartments. We seeded 10 infections in Lusaka District where the capital city is located (10 to avoid too many stochastic fadeouts). We ran 100 simulations for 1 year to capture the early dynamics of an outbreak across Zambia. Details of the transmission model and initial parameters are provided in S3 Text.

For each mobility matrix derived from the various data sources and their combinations, we calculated the proportion of districts with an introduction event, defined as having the mean number of cumulative infections greater than or equal to one. We also calculated the mean number of cumulative cases over the course of 1 year and the 2.5th and 97.5th quantiles of cumulative cases nationwide.

As sensitivity analysis, we conducted simulations of measles virus dynamics following the introduction of 5 cases each in Ndola District in northern Zambia and Choma District in southern Zambia. We used 5 cases in each of the two districts to seed the same number of total infections as in the main scenario (10) but distributed between the two districts due to their lower connectivity compared to Lusaka District.

## Identifying what proportion of the population would have to be described by the travel survey to result in different estimates of introduction events than using mobile phone data to quantify population mobility

In this approach, a portion of the population had mobility patterns described by the mobile phone data and the remaining proportion of the population described by the travel survey. We selected these two data sets as mobile phone data had

the highest spatial coverage in Zambia while the travel survey had the lowest. We scaled the standard deviation on travel survey-fitted coefficients so that precision relative to the mean was equivalent to that of the mobile phone data-fitted coefficients. We varied the proportion of population described by mobile phone data from 0 to 1, obtained a weighted average of the mobile phone-informed and travel survey-informed mobility matrices, and carried out simulations as described above using the weighted mobility matrix. Simulations were repeated 100 times and the proportion of districts with introduction events was obtained as previously described.

## Ethics statement

This study was submitted to the Johns Hopkins Bloomberg School of Public Health Institutional Review Board and was found to be Not Human Subjects Research (FWA #00000287), as it relies on use of existing, de-identified data collected prior to this study. Namely, the 2018 ZDHS data are publicly available (https://microdata.worldbank.org/index.php/catalog/3597). The ethics application for secondary data analysis for characterizing mobility data gaps, using different sources including the Data for Good at Meta, has been also approved by the Ethics Committee of the Faculty of Environmental and Life Science at the University of Southampton (ERGO II 87924). Collection and analysis method complied with the terms and conditions for the source of the data (Data for Good at Meta). The travel survey data were collected as a part of a serology study in Ndola and Choma Districts (human subjects research approval from Johns Hopkins Bloomberg School of Public Health IRB (IRB00018265), National Health Research Authority, Zambia (IRB00002911), and the Tropical Diseases Research Centre Ethics Review Committee (FWA0003729)). All participants 18 years and over in the study provided informed consent; for individuals under 18 years old, parents or guardians provided parental permission for the child. Written consent was obtained where possible; otherwise, a thumbprint was collected. In addition, verbal assent was obtained and documented for children 10 – 17 years old. Consent, assent, and parental permission forms were available in English, Bemba, and Tonga, and were administered in the language of choice of participants.

## Results

### Data sets measured different components of individual and population-level mobility

We analyzed four types of mobility data: community-based travel survey from two districts (2022), national multi-stage sampled DHS (2018), changes in locations of Facebook application users (4/2020–5/2022), and mobile phone data from a leading operator (3/2020-12/2020) (see Materials and Methods). The two individual-level data sets (travel survey and DHS) collected responses about overnight travel in the past two months (travel survey) and 12 months (DHS), but only the travel survey included the destination location. As the DHS interviewed women aged 15–49 years and men aged 15–59 years, the median age of respondents was 27 years old (IQR [20; 37]), with 53% of respondents women. The travel survey used stratified sampling to enroll caregivers of young and older children, as well as other adults in the home at the time of the survey; as a result, the sample was predominantly female (83%), with slightly older participants (median age 34 years; IQR [26; 43]; maximum age 96 years). The two population-level data sets (mobile phone data and Facebook mobility data) both included spatially and temporally aggregated flows between districts, with no information available about individuals in the data sets (see S1 Table).

The mean probability of departure, calculated on a monthly scale for all data sets except Facebook (see Materials and Methods), was similar across data sets, ranging from 4% (DHS and travel survey) to 8% (mobile phone and Facebook data). The mobile phone and Facebook data sets were the most strongly correlated (Spearman's correlation coefficient 0.529, p-value = 0.045) (Fig 1C, S2A Fig, S2 Table). The probability of departure was generally highest in mobile phone data (median = 5.8%, IQR [4.2%; 8.4%]), followed by Facebook data (median = 4.5%, IQR [0.6%; 21.8%]), travel survey (3.7% and 4.3%), and DHS (median = 3.8%; IQR [3.0%; 4.3%]) (S2B-C Fig). The probability of departure was 0 for most districts in the Facebook data set, notably for all the least populated districts (<112,000 people in 2020, 48/115 districts) and were excluded from additional analyses (S1B Fig).

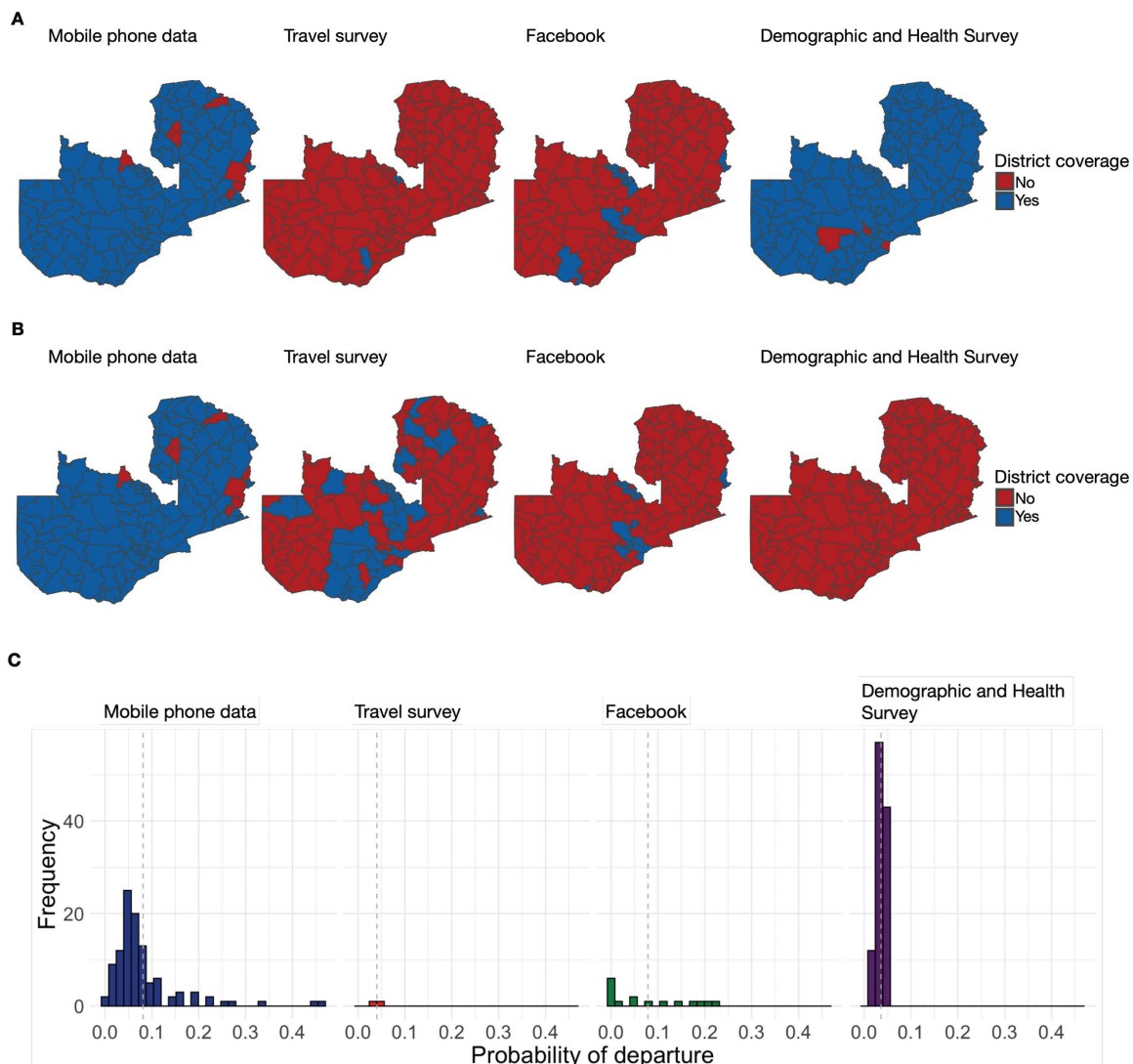

**Fig 1. Coverage of the origin and destination districts in mobility data sets and the probability of travel from districts. A.** Geographic coverage of districts of origin for each of four mobility data sets included in analysis. A district is said to be covered if there was a non-zero probability of travel reported from that district. **B.** Geographic coverage of destination districts for each of four mobility data sets included in analysis. A district is said to be covered if there were a non-zero number of trips reported into that district from another district. Destination information was not collected in the Demographic and Health Survey. **C.** Histogram of district-level probabilities of departure (travel) out of districts. Dashed line represents the mean probability of departure observed in each mobility data set. Maps were created in R, using base layer shapefile provided by OCHA under Creative Commons Attribution for Intergovernmental Organisations license [37].

The largest range of differences across data sources was in spatial coverage. For example, there were between 2 – 112 (2% - 97%) districts represented as origin locations for which the probability of departure was quantified (DHS = 112, mobile phone data = 107, Facebook data = 16, travel survey = 2) (Fig 1A) and between 0 – 107 as destination locations (DHS = 0 (no information on destination collected), mobile phone data = 107, Facebook data = 19, travel survey = 39) (see Fig 1B; S1 Fig).

## Adjusting for heterogeneity in the probability of overnight travel by mobile phone ownership and internet use from the DHS resulted in a decrease in the probability of departure in mobile phone and Facebook data

To assess the potential for sampling bias, based on mobile phone ownership for the mobile phone data and internet usage for the Facebook data set, we explored heterogeneity in travel by mobile phone ownership and internet use as reported in the DHS. The district-level probability of overnight travel in the DHS was lower for individuals who did not own a mobile phone or did not use the internet (Fig 2A-B) compared to mobile phone owners. Adjusting for ownership (mobile phone) and internet use (Facebook) biases from the DHS data resulted in a lower estimated probability of departure in 94% of districts for both Facebook and mobile phone data. The median change in probability of departure between unweighted and weighted estimates was -12% for mobile phone data (IQR [-22%; -3%]), and -39% for Facebook data (IQR [-100%; -8%]) (Fig 2C). Notably, weighting reduced the probability of departure to 0 for some of the districts (-100% change).

## Pooling probabilities of departure allowed for combined information on departure across data sets

Given the spatial sparsity of data on departure probabilities for the travel survey and Facebook data, we used pooling to obtain a single estimate of departure for each district using all four data sets. We first fit a beta-binomial model for the probability of departure from districts with available data and used this model to predict the probability of departure for

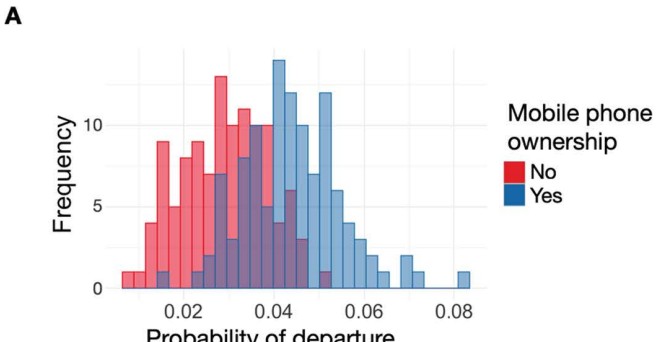

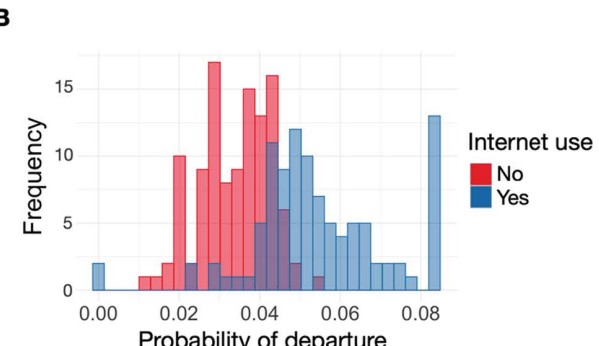

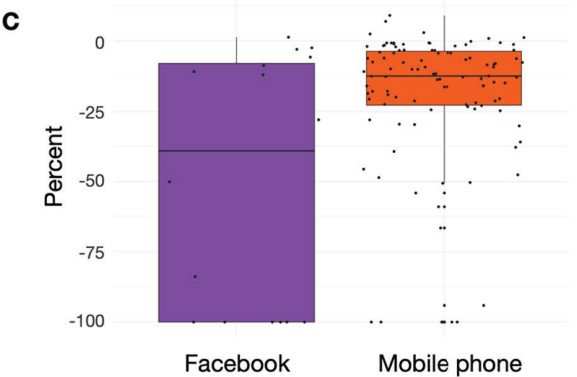

**Fig 2. Accounting for heterogeneity in the probability of travel by mobile phone ownership and internet use in the last 12 months. A.** Histogram of district-level probabilities of departure in the Demographic and Health Survey, stratified by reported mobile phone ownership. **B.** Histogram of district-level probabilities of travel in the Demographic and Health Survey, by district, stratified by reported internet use in the last 12 months. **C.** Percent change in probability of travel for each district following weighting by heterogeneity in travel by mobile phone ownership (mobile phone data set) and internet use (Facebook), as measured in the Demographic and Health Survey. Each point represents a district.

all 115 districts in Zambia, regardless of coverage by each data set. We calculated pooled estimates of the probability of departure for each district using a random effect and inverse probability weighting (see Materials and Methods). Compared to individual data set estimates, the pooled estimates generally had smaller variance than Facebook data and travel survey estimates, except for a few districts where the mobile phone estimate greatly differed from the DHS and Facebook estimate (see S3A-B Fig). The pooled estimates also consistently decreased the departure probability compared to the raw mobile phone or Facebook data (S3A-B Fig). Pooling weighted mobile phone and Facebook data sets alongside DHS and travel surveys consistently resulted in a lower probability of departure than pooling unweighted data (S3C Fig).

**Distance played a smaller role in the diffusion process estimated from the travel survey although estimates from the travel survey were less precise**

Using an exponential gravity model fit to each data set, we estimated travel across Zambia (see S1 Text for goodness of fit metrics from a range of spatial interaction models). Parameter estimates were consistent between the mobile phone and Facebook data sets with only the travel survey suggesting a considerably different parameter for the lower probability of travel with increasing distance (distance deterrent parameter). That is, from travel survey data, travel between two locations diminishes at a slower rate as distance increases, compared to other data sets. To leverage information across data sets simultaneously, we fit combined Bayesian exponential gravity models where the coefficient estimates from a model fit to one data set were used as a prior in the fit for a model from another data set. In general, given the imprecise nature of the fitted coefficients for the travel survey data, using this information as a prior did not substantially change the posterior distributions of model coefficients from either the mobile phone or Facebook data sets. The only exceptions were the two population size parameters, where using either travel survey or mobile phone data as a prior decreased the posterior estimates from the Facebook data (Fig 3). Conversely, models fit to travel survey data were sensitive to using either Facebook or mobile phone data as priors. The model coefficients fit to mobile phone data were more precise and, as a result, using Facebook coefficients as a prior did not shift the posterior (Fig 3). In these combined approaches, a single data set would need to be substantially more precise to serve as an informative prior when using data sets with higher spatial coverage (e.g., mobile phone data) (see S4 Fig).

**Assessing the robustness of measles simulations to different estimates of mobility**

Using a stochastic compartmental model, we simulated the introduction of measles cases into the most populated, capital district Lusaka District (see S5 Fig for sensitivity analysis with varying initially infected locations), with mobility between districts parameterized by different combinations of departure and diffusion processes (see Materials and Methods). Although the final epidemic size was relatively similar (Fig 4A, S3 Table), when a single data set (travel survey, Facebook, or mobile phone data) was used to inform departure and diffusion processes, there were marked differences in the proportion of districts with at least one introduction event. Using mobile phone data only resulted in 35% of districts experiencing introductions, compared to 42% of districts using Facebook data and 98% of districts using travel survey data (S3 Table, Fig 4B). The relative rankings of whether an introduction would occur in a specific district was consistent across data sets (correlation range 0.76-0.91, S4 Table), but ranks for the mean time to introduction were less correlated (correlation range 0.22-0.68). While Facebook and mobile phone data were still moderately correlated (Spearman's rho = 0.68), the correlation between travel survey results and either Facebook or mobile phone data results was low (S4 Table).

We next assessed whether combining information on diffusion and departure processes across combinations of data sets and pooled estimates affected simulated measles virus transmission dynamics. The probability of introduction remained below 40% if mobile phone data were used to inform the diffusion process, regardless of which data set was used to inform departure (weighted mobile phone data, DHS, pooled weighted or pooled raw) (Fig 4C, S3 Table). When Facebook data were used to inform diffusion, however, the probability of introduction was more sensitive to the choice of the departure data set, falling below 40% when DHS data or pooled estimates were used (S3 Table, Fig 4C). When diffusion was informed by

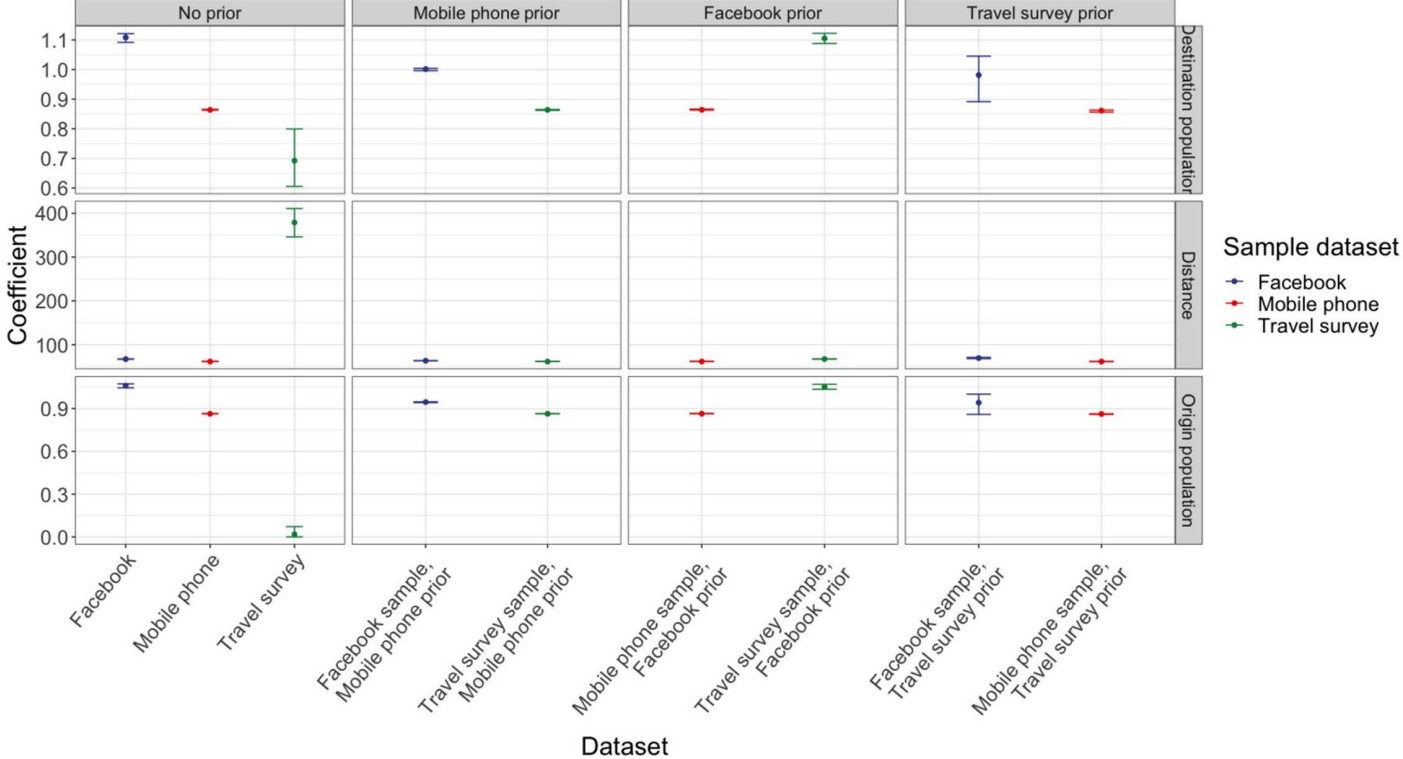

**Fig 3. The coefficient estimates for distance deterrence, origin population size, and destination population size parameters, using different combinations of sample data sets and priors.** Distance deterrence parameter indicates the extent to which probability of travel decreases with increase in distance between origin and destination locations. Results are split into columns by the prior used (no prior, mobile phone data set prior, Facebook prior, and travel survey prior), and color-coded by which data set was used as a sample. The panel in the top-most left quadrant shows estimates for destination population coefficient when no prior data set was used; i.e., mobility model was fit to a single data set at a time (Facebook, mobile phone, or travel survey). The topmost panel in the second column shows estimates for the same coefficient (destination population) when a model was fit using mobile phone data as a prior, and either Facebook data or travel survey as the sample.

the travel survey data, the results were most sensitive to the choice of departure data set, with less than 40% of districts having introductions when pooled estimates were used (S3 Table, Fig 4C). Notably, less than 40% of districts had introduction events when pooled estimates were used for departure, regardless of which data set was used to inform the diffusion process (S3 Table, Fig 4C). Use of travel survey data as a prior and Facebook data to inform likelihood did not change the proportion of districts with introduction events compared to using an uninformative prior (Fig 4C).

Finally, given the predominant use of a single data set to quantify mobility, we considered a population composed of one subgroup whose mobility had travel patterns described by the mobile phone data and a second subgroup whose mobility was best described by the travel survey. At least half of the population would have to be in the second subpopulation for the proportion of districts with introduction events to change by at least 25%. However, confidence intervals on the proportion of districts with introduction events increased as the proportion of the subpopulation with mobility described by the travel survey also increased (S6 Fig).

## Discussion

As more data quantifying human mobility become available, it is increasingly important to understand the biases and limitations of these data sets to capture the processes of departure and diffusion. Inconsistency in mobility measures

**A**

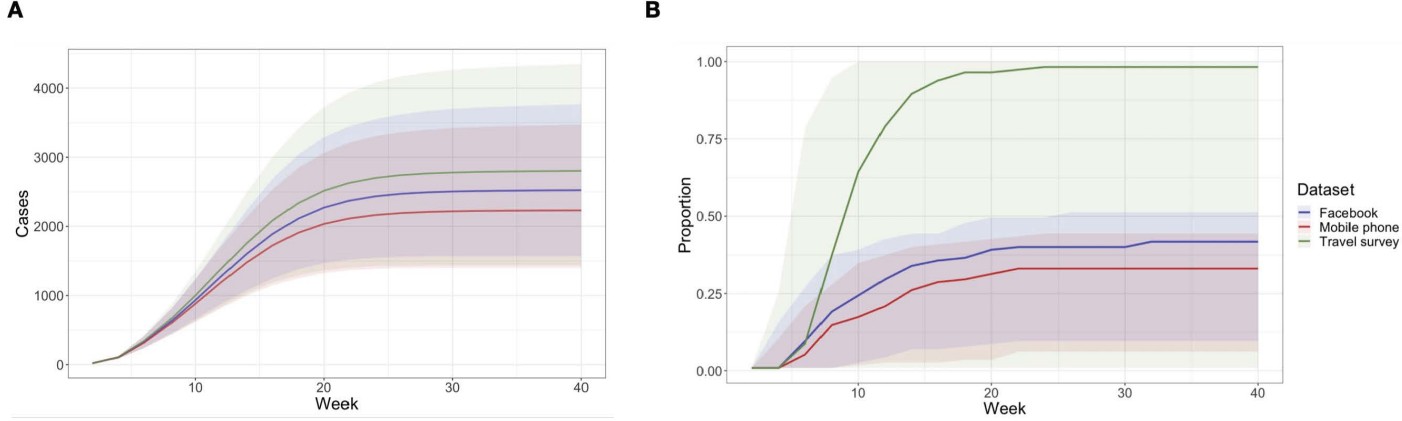

**C**

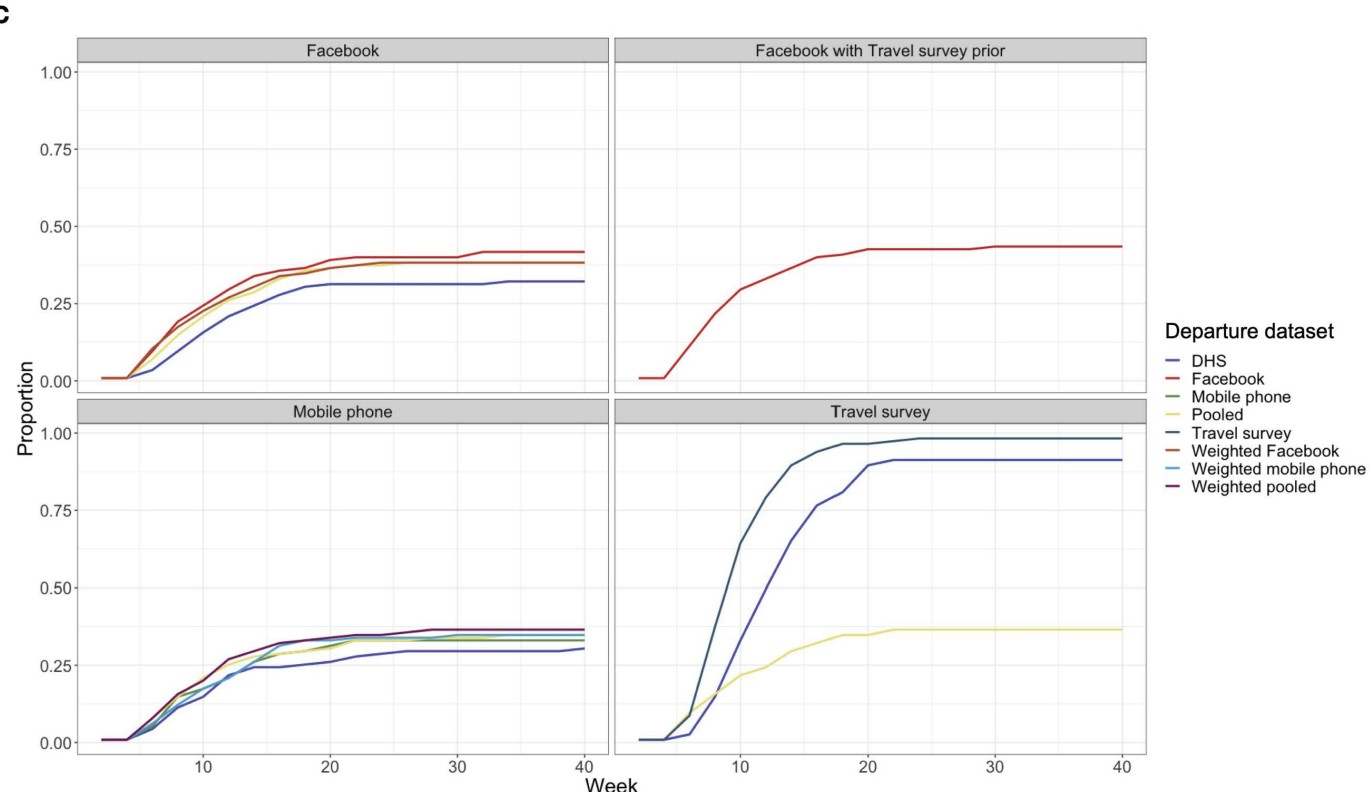

**Fig 4. Results of simulations of measles transmission dynamics following the introduction of 10 infectious cases into Lusaka District. A.** Cumulative cases of measles with mobility between districts informed by a single data set (mobile phone, Facebook, or travel survey). **B.** Proportion of districts with introduction events with mobility between districts informed by a single data set. **C.** Proportion of districts with introduction events with results split by which data set was used to inform the diffusion process.

obtained from different data sources may have significant implications on infectious disease transmission models, especially as these models are used to classify spatial risk for diseases like measles. We compared mobility measures across Zambia from four commonly used data sets and evaluated the impact of using these data sets on estimates of simulated measles virus transmission across the country. The four data sets we examined - DHS, travel survey, mobile phone data,

and Facebook data – differed in terms of geographic coverage, population demographics, and type of movement quantified. Both the departure and diffusion processes varied widely across data sets, with mobile phone and Facebook data generally estimating a higher proportion of travel out of district compared to DHS and travel survey data. Using these data sets in a measles virus transmission model resulted in different estimates of the proportion of districts with measles virus introduction events across the country, suggesting that the choice of mobility data set has significant implications for informing disease control programs. Most strikingly, the use of the travel survey to quantify mobility resulted in 98% of districts with measles introduction events. This would provide strong evidence for a nationwide measles campaign, rather than selected supplementary immunization activities. However, if mobile phone or Facebook data were used to quantify mobility, the proportion of districts with introduction events fell below 50%, suggesting that supplementary immunization activities could target a smaller portion of the country. While this study was carried out in Zambia, the implications of these findings are applicable to other settings and phases of public health response. Since mobility data are increasingly being used to inform decision making before, during, and after a disease outbreak, understanding how various data sets can be used to produce realistic estimates of spatial pathogen dynamics is needed. The choice of data set can impact estimates of introduction risk which would ultimately influence the scale and focus of public health interventions.

The range in spatial coverage of each data set made it difficult to perform systematic comparisons of how they differed and methods to combine data sets. While we modeled mobility using available data, and used predicted mobility for geographical areas where data were missing, this relied on the assumption that mobility data were missing at random. The time frame during which mobility was measured also varied, with DHS taking place before the COVID-19 pandemic. However, other studies show that while mobility in Zambia decreased during periods of travel restrictions, it returned to baseline levels within a few months of the start of the pandemic [38]. The differences in the data can make it difficult to use one data set in lieu of another, particularly during an emergency period. Due to the much larger number of locations represented in the mobile phone data, this data set dominated the posterior in the Bayesian approach to modeling the diffusion process. On the other hand, the coefficients estimated from fitting the mobility models to travel survey data would have to be much more precise to serve as an informative prior when combined with mobile phone or Facebook data. This level of precision may not be reachable by increasing the sample size of the travel survey alone and, given the large disparities between these data sets, understanding how informative travel captured by the travel survey compares to other data sets with wider spatial coverage would need to be determined. When no information about the relative importance of data sets exists, pooling estimates across data sets can be a useful option. However, this may result in lower precision and obfuscate some of the fundamental differences in types of mobility and population captured in each of the mobility data sets.

The observed differences in estimates of both departure and diffusion processes across data sets included in this study are concerning, especially as a single data set is usually used to quantify mobility in models of disease transmission. The use of mobile phone and Facebook data resulted in a highly correlated ranking of districts based on probabilities of introduction, which could provide confidence in the allocation of resources to different districts. However, the correlation was not consistently high for ranks of the introduction time.

While data sets like mobile phone data have high geographical coverage, capture variation in travel over time, and can be made available almost in real-time, additional studies must be undertaken to quantify travel differences among individuals included in mobile phone data and those who are excluded. Existing literature provides evidence that mobile phone ownership differs by socio-economic and demographic factors like age, gender, and other characteristics [39–43]. Previous analysis using the travel survey and mobile phone data used in this study found that travel varied by mobile phone ownership and by age [26]; with young children less likely to own or carry mobile phones, their exclusion from mobile phone data is likely to result in biased estimates of travel. Furthermore, even among individuals that do own a mobile phone, financial restraints in making calls or sending a message [44], especially in presence of alternative options like WhatsApp [45], may result in exclusion from call detail records. On the other hand, travel recorded by a unique SIM card may not directly map to travel by a single individual. Some individuals may own multiple SIM cards; others may travel with

children or other individuals that do not own a mobile phone or a SIM card from the same network provider. For example, in the travel survey, caregivers of children under 5 years of age reported that when they undertook an overnight trip, they traveled with on average 0.48 children in Choma and 0.59 children in Ndola [26]. SIM cards used in Internet of Things devices may also be captured in call detail records but would capture a different dimension of travel than SIM cards in mobile phones, especially if used in long-haul vehicles. The extent to which this could bias departure and diffusion processes captured by mobile phone data is likely to vary across settings.

In addition to sampling biases, mobility data collected using digital devices like mobile phones may have varying accuracy depending on devices and characteristics of networks used in data collection. For instance, the area covered by a cell tower in areas with 2G network are larger than areas covered by 3G, 4G, and 5G networks. With 2G more common in rural areas of Zambia, it is likely that there was higher inaccuracy in travel origin and/or destinations when travel included at least one terminus in a rural area, potentially resulting in higher rate of misclassification of district of origin or destination.

Facebook data are similarly subject to sampling biases as well as issues with accuracy of location. Increasing use of multiple Global Navigation Satellite Systems as opposed to GPS provides a promising direction for increasing accuracy of geo-location on a large scale [46]. Individuals that do not use a smartphone, do not have Facebook application installed, or choose not to opt in to location sharing on their application are excluded from the data set. To mitigate the issue of sampling bias in mobile phone and Facebook data, we used information available in DHS to adjust mobility estimates, accounting for heterogeneity in travel outside district for individuals with and without mobile phones, and reporting or not reporting Internet use in the last 12 months. However, as these questions capture a different facet of mobility and a different time period, there likely remains residual bias. Furthermore, to protect privacy and confidentiality of its users, the data set omits information on routes between destination and origin tiles that have less than ten users reporting travel; as a result, Facebook data are likely to bias models towards higher-flow routes.

Capturing mobility via a travel survey faces a different set of challenges. Previous studies have shown that survey-based reporting of travel may differ by trip purpose [47], and may underreport frequency of travel while over-reporting long-distance trips [48]. The latter could explain why the distance deterrent coefficient was much smaller in the travel survey than in mobile phone and Facebook data, which could in turn explain why use of travel survey findings in simulations of measles transmission resulted in introduction events across the country. If individuals are less likely to report shorter, more routine trips, we are likely to overestimate long-distance travel, resulting in an inflated probability of departure to farther districts. Furthermore, the relatively small sample size of surveys may result in capturing of the high-volume travel routes but is less likely to capture smaller-frequency connections between origin and destination districts. In addition, the travel survey analyzed in this study included a recall period of two months prior to beginning of the survey and was administered over a three-month period. If there was significant variation in travel over the course of the year, for example seasonal travel or travel during holidays, the fluctuation may be missed.

Ultimately, more refined estimates of human mobility can improve our understanding of spatial patterns of disease transmission and allow for shaping more effective disease control programs. Increased availability of mobility data through sources like GPS trackers, mobile phone, social media, satellite data, retail and commercial data, and others offer a unique opportunity to refine and produce more accurate estimates of mobility. However, further research on ways to correct for sampling and information biases inherent in each of the data sets, and implications of these corrections on public health inference, is necessary. While few data sets make available socio-economic and meta-data of individuals included in the sample, stronger efforts should be put to ascertaining this information and using it to correct for known biases, especially in the sampling processes. Furthermore, despite the increasing availability of multiple mobility data sets, the majority of research relies on a single data set to quantify mobility [23]. Identifying appropriate ways to combine information from multiple data sources, while accounting for difference in populations included in each as well as heterogeneity in mobility metrics captured, is crucial. In the meantime, nationwide surveys like DHS or the Zambia Population-based HIV/AIDS Impact Assessment could integrate a travel survey module to provide a representative sample of the population,

such as the movement of children, who are missed from many other data sets but are important to the transmission of many pathogens. Additionally, metadata included in population-level data sets, such as gender, age, and socio-economic status of individuals for whom data were collected, could help quantify the direction and magnitude of bias. Leveraging the information available across different sources of mobility data has the potential to adjust parameter estimates, improve geospatial and temporal coverage of data, and better capture travel patterns of populations.

## Supporting information

**S1 Table. Key features of datasets used to estimate population mobility in Zambia.**
(DOCX)

**S2 Table. Spearman's rho correlation for probabilities of departure between three datasets used to quantify mobility, Zambia.** Travel survey is excluded from this comparison given that it had information on travel from only two origin districts (Ndola and Choma Districts).
(DOCX)

**S3 Table. Results of measles simulations, by the source of departure and diffusion mobility data.** Cumulative infections show mean and 95% confidence interval values of cumulative infections over the course of simulation time (72 weeks). Peak infections are the mean and 95% confidence interval for the maximum number of infections in a single 2-week timestep.
(DOCX)

**S4 Table. Spearman's rho correlation between ranks in district-level probability of introduction events, and time to introduction events, depending on the dataset used to inform mobility between districts.**
(DOCX)

**S1 Text. Modeling diffusion processes.** Availability of data on diffusion, fitting an exponential gravity model, assessing goodness of fit.
(DOCX)

**S2 Text. Incorporating information from priors and observed data using Bayesian framework.**
(DOCX)

**S3 Text. Transmission model specification.**
(DOCX)

**S1 Fig. Origin-destination matrices for three datasets with information on diffusion processes in Zambia.**
A. Mobile phone data, B. Facebook dataset, C. Travel survey.
(DOCX)

**S2 Fig. Observed probabilities of departure from district from Mobile phone dataset, Facebook, Demographic and Health Survey (DHS), and Travel survey.** A. Boxplots of district-level probabilities of departure, by province. Outliers have been removed from the plot to facilitate visibility. The travel survey was only done in two districts, in Copperbelt and Southern provinces, respectively; no data are available from the survey in other districts and provinces. Facebook data were only available for districts in four provinces. B. The probability of travel by district, in selected districts. Choma and Ndola were included as the only districts with Travel survey data. Other districts are the five districts with the largest difference in probability of departure across datasets. Observed probabilities of departure from district from Mobile phone dataset, Facebook, Demographic and Health Survey (DHS), and Travel survey. C. All observed probabilities of departures for the four datasets.
(DOCX)

**S3 Fig. Probability of travel out of the district, comparing estimates from the beta-binomial model fit to raw datasets, and pooled estimates.** A. Probability of travel, where pooled value was obtained by pooling estimates from raw Mobile phone data, Demographic and Health Survey (DHS), Facebook, and Travel survey. B. Probability of travel, where pooled values were obtained by pooling estimates from DHS and Travel survey, and weighted Mobile phone and Facebook data. C. Comparison of probabilities of travel using pooled estimates from raw datasets (Mobile phone data, Facebook, Travel survey, and Demographic and Health survey) and the mixture of raw and weighted datasets (weighted Mobile phone data, weighted Facebook, Travel survey, and Demographic and Health survey). Each point represents a district. X-axis is the probability of leaving from pooled estimates from values obtained through fitting the beta-binomial model to raw datasets. The diagonal line indicates a boundary of no change in probabilities.
(DOCX)

**S4 Fig. Proportion of districts in Zambia with measles introduction events, as observed in simulations of measles transmission dynamics with population composed of a sub-population whose movement patterns are captured by mobile phone data, and sub-population whose movement patterns are captured by travel survey.**
(DOCX)

**S5 Fig. Results of sensitivity analysis: simulations of measles dynamics following the introduction of 5 cases into Ndola district and 5 cases into Choma district.** A. Cumulative measles cases in scenarios where a single data set was used to inform departure and diffusion processes. B. Proportion of districts with introduction events, with mobility between districts informed by different combinations of datasets. C. Proportion of districts with introductions, with panels representing dataset used to inform diffusion process. D. Cumulative cases, with panels representing dataset used to inform diffusion process. E. Mean cumulative cases of measles from simulation, with results split by which dataset was used to inform diffusion process.
(DOCX)

**S6 Fig. Posterior estimates of coefficients in exponential gravity model, using travel survey-fitted coefficients as a prior and mobile phone data (A) and Facebook data (B) as likelihood.** In both panels, the x-axis is the scaling factor by which the standard deviation of the coefficients of mobility model fit to travel survey was reduced by.
(DOCX)

## Acknowledgments

The authors would like to acknowledge the Data for Good program at Meta for sharing data and thank Dr. Fatumah Atuhaire for Facebook mobility data collation.

## Author contributions

**Conceptualization:** Natalya Kostandova, Bryan Lau, Amy Wesolowski.

**Data curation:** Natalya Kostandova, Christine Prosperi, Simon Mutembo, Chola Nakazwe, Harriet Namukoko, Bertha Nachinga, Shengjie Lai, Andrew J Tatem, Qianwen Duan, Elliot N Kabalo, Kabondo Makungo, Gershom Chongwe, Innocent Chilumba, Gloria Musukwa, Kalumbu H Matakala, Mutinta Hamahuwa, Webster Mufwambi, Japhet Matoba, Irene Mutale, Kenny Situtu, Edgar Simulundu, Phillimon Ndubani, Alvira Z Hasan, Shaun A Truelove, Amy K Winter, Andrea C Carcelen.

**Formal analysis:** Natalya Kostandova.

**Investigation:** Natalya Kostandova, Andrew J Tatem, Gershom Chongwe, Edgar Simulundu, Phillimon Ndubani.

**Methodology:** Natalya Kostandova, Bryan Lau, William J Moss, Amy Wesolowski.

**Project administration:** Simon Mutembo, Amy Wesolowski.

**Resources:** William J Moss, Amy Wesolowski.

**Software:** Natalya Kostandova.

**Supervision:** Natalya Kostandova, Christine Prosperi, Bryan Lau, Amy Wesolowski.

**Validation:** Natalya Kostandova, Christine Prosperi, Shengjie Lai, Bryan Lau, William J Moss, Amy Wesolowski.

**Visualization:** Natalya Kostandova.

**Writing – original draft:** Natalya Kostandova.

**Writing – review & editing:** Christine Prosperi, Shengjie Lai, Andrew J Tatem, Qianwen Duan, Kabondo Makungo, Gershom Chongwe, Innocent Chilumba, Mutinta Hamahuwa, Japhet Matoba, Edgar Simulundu, Phillimon Ndubani, Amy K Winter, Andrea C Carcelen, Bryan Lau, William J Moss, Amy Wesolowski.

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
