## [Decision Letter · Decision Letter 0]

3 Sep 2024

PGPH-D-24-01677

Comparing different sources of human mobility data and methods of integration: modeling measles virus transmission in Zambia

Dear Natalya,

Thank you for submitting your manuscript to PLOS Global Public Health. After careful consideration, we feel that it has merit but does not fully meet PLOS Global Public Health’s publication criteria as it currently stands. Therefore, we invite you to submit a revised version of the manuscript that addresses the points raised during the review process.

We look forward to receiving your revised manuscript.

Kind regards,

Collins Otieno Asweto, PhD

Academic Editor

Journal Requirements:

1. In your Methods section, please include additional information about your dataset and ensure that you have included a statement specifying whether the collection and analysis method complied with the terms and conditions for the source of the data.

Reviewer's Responses to Questions

**Comments to the Author**

1. Does this manuscript meet PLOS Global Public Health’s publication criteria ? Is the manuscript technically sound, and do the data support the conclusions? The manuscript must describe methodologically and ethically rigorous research with conclusions that are appropriately drawn based on the data presented.

Reviewer #1: Partly

Reviewer #2: Yes

2. Has the statistical analysis been performed appropriately and rigorously?

Reviewer #1: Yes

Reviewer #2: Yes

3. Have the authors made all data underlying the findings in their manuscript fully available (please refer to the Data Availability Statement at the start of the manuscript PDF file)?

Reviewer #1: Yes

Reviewer #2: Yes

4. Is the manuscript presented in an intelligible fashion and written in standard English?

Reviewer #1: Yes

Reviewer #2: Yes

5. Review Comments to the Author

Reviewer #1: The manuscript titled, "Comparing different sources of human mobility data and methods of integration: modelling measles virus transmission in Zambia" is a credible innovative analysis. It however requires some minor review to meet PLOS Global Public Health's standards. The following major comments about the manuscript are stated below:

1. The manuscript lacks clear objective (the objective is expected to be SMART). Although it was written at the end of the introduction, it looks cumbersome and needs to be reviewed.

2. The method of comparison and the automated meta-analysis of data needs to be clearly provided in the method section.

3. Some descriptions of methods were found in the results section, and this is not acceptable.

4. Some discussions were also included in the results section, and this is not in line with the manuscript's guideline

5. The Method section is supposed to come before the results.

If these observed comments were considered by the authors, I would suggest that the manuscript should be considered for publication.

Reviewer #2: The manuscript titled Comparing Different Sources of Human Mobility Data and Methods of Integration: Modeling Measles Virus Transmission in Zambia presents a significant effort in analyzing the integration of human mobility data using a Bayesian framework to model measles transmission dynamics in Zambia. The study is innovative however following points should be addressed in the revised version to enhance the manuscript’s clarity, precision, and impact.

The introduction, while informative, requires strengthening to clarify the specific objectives of the study. The opening sentence referencing the first 16 citations lacks specificity regarding the modes of transmission studied, which range from fecal-oral to airborne and vector-borne. Providing these details would immediately set the stage for the relevance of the study within the broader context of infectious disease transmission modelling. Furthermore, recent studies that explore the use of human mobility data in the context of SARS-CoV-2, such as Berke A, Doorley R, Alonso L, Arroyo V, Pons M, Larson K. Using mobile phone data to estimate dynamic population changes and improve the understanding of a pandemic: A case study in Andorra. PLoS One. 2022 Apr 26;17(4):e0264860. doi: 10.1371/journal.pone.0264860, should be incorporated to reflect the most current advancements in the field. This would not only broaden the context but also underscore the applicability of the study's findings to contemporary public health challenges.

The manuscript’s reference to the Global Positioning System (GPS) should be more precise, particularly in distinguishing GPS from the broader category of Global Navigation Satellite Systems (GNSS). While the term GPS is initially used correctly within the manuscript, later uses are confusing. It is essential to acknowledge that modern devices often utilize multiple GNSS systems, such as Galileo or GLONASS beyond GPS. This distinction is critical for accuracy, as it affects the interpretation of the data and its generalizability to environments with varying GNSS coverage, especially in light of ever-increasing GPS Spoofing. (A. Altaweel, H. Mukkath and I. Kamel, "GPS Spoofing Attacks in FANETs: A Systematic Literature Review," in IEEE Access, vol. 11, pp. 55233-55280, 2023, doi: 10.1109/ACCESS.2023.3281731)

On line 73, the limitations of call data should be more explicitly discussed. Studies have shown that in many regions, VoIP services like WhatsApp have overtaken traditional mobile phone usage due to the lower cost of data compared to voice calls. This shift can significantly impact the representativeness of the data. Studies have demonstrated that reliance on phone call data might omit substantial segments of the population who primarily use VoIP or have financial restraints in making mobile phone calls (doi: 10.1016/j.socscimed.2018.01.011 and doi: 10.1177/08944393221111340)

The discussion starting at line 89 would benefit greatly from an in-depth review of prior work, particularly studies that have integrated multiple data sources for infectious disease modeling. (doi: 10.1080/17538947.2021.1886358) This would not only situate your work within the existing literature but also highlight the innovative aspects of your approach.

Line 289 and the subsequent discussion sections should explicitly address the limitations of the study, particularly concerning the mobile phone data. For instance, during the data collection period, Zamtel was operating on a 2G network in some regions, which may have affected the accuracy of location data due to the broader cell tower range in flatter terrains. This could result in devices being incorrectly located in a different district than where they actually were. A discussion of this limitation is necessary to temper the interpretation of the results.

Furthermore, on line 315, it is important to acknowledge that a unique SIM card does not necessarily correspond to a mobile phone. Many IoT devices, such as vehicle trackers, also utilize SIM cards, which could skew the data if not accounted for. If the data set used in the study was filtered to include only mobile phone users, this should be clearly stated to avoid any potential misinterpretation.

The Materials and Methods section would benefit from a more detailed discussion of the limitations of the various data sources used. For instance, the potential for recall bias in the travel survey data is significant and should be addressed. Additionally, the explanation of the Bayesian framework should be expanded. It is currently not perfectly clear how priors were selected and how they influenced the posterior distributions, which is particularly important for readers who may not be well-versed in Bayesian methodologies. The justification for the random-effects approach, although sound, could also be expanded with reference to foundational statistical texts, such as Gelman et al. (2004) (doi: 10.1214/009053604000001048), to provide a more rigorous defense of this choice.

The brief mention of the SARS-CoV-2 pandemic’s impact on mobility patterns requires further elaboration. Specifically, a discussion on how the pandemic may have affected the validity of the data and whether any novel methods could account for these sudden and drastic changes would add substantial value to the manuscript. This could include, for example, the potential for real-time data correction methods or the use of alternative data sources during periods of lockdown or reduced mobility.

The choice of a beta-binomial model is appropriate, but the manuscript lacks detail on the hierarchical structure of the model and its implications. Were the district-level effects modeled as random effects, and if so, how were these distributed? This is an important detail that influences the interpretation of the model’s results. Additionally, the discussion should include the potential trade-offs involved in pooling data, such as the risk of introducing biases or obfuscating important variations that are specific to individual data sets.

There is also mention of districts with zero probabilities in departure indicating outliers or sparse data points. However, it should be clarified whether specific methodologies such as regularization were utilized in the modelling process in handling of such data.

The innovative aspects of this study should be highlighted more prominently, particularly in the context of the literature review. Additionally, the discussion and conclusion sections should be expanded to ensure a more logical flow and thorough exploration of the study’s implications. The conclusions drawn from the study should be clearly linked back to the objectives stated in the introduction, providing a cohesive narrative that underscores the study’s contributions to the field of infectious disease modeling.

By addressing these points, the authors can significantly enhance the manuscript’s clarity, methodological rigor, and overall impact. I recommend revising the manuscript to incorporate these critical elements and re-submitting it for further review.

6. PLOS authors have the option to publish the peer review history of their article (what does this mean? ). If published, this will include your full peer review and any attached files.

**Do you want your identity to be public for this peer review?** For information about this choice, including consent withdrawal, please see our Privacy Policy .

Reviewer #1: **Yes: ** Sikiru Olanrewaju Badaru

Reviewer #2: **Yes: ** Baran Erdik. MD. MHPA

---

## [Decision Letter · Decision Letter 1]

15 Nov 2024

PGPH-D-24-01677R1

Comparing different sources of human mobility data and methods of integration: modeling measles virus transmission in Zambia

Dear Natalya,

Thank you for submitting your manuscript to PLOS Global Public Health. After careful consideration, we feel that it has merit but does not fully meet PLOS Global Public Health’s publication criteria as it currently stands. Therefore, we invite you to submit a revised version of the manuscript that addresses the points raised during the review process.

We look forward to receiving your revised manuscript.

Kind regards,

Collins Otieno Asweto, PhD

Academic Editor

Journal Requirements:

Reviewer's Responses to Questions

**Comments to the Author**

1. If the authors have adequately addressed your comments raised in a previous round of review and you feel that this manuscript is now acceptable for publication, you may indicate that here to bypass the “Comments to the Author” section, enter your conflict of interest statement in the “Confidential to Editor” section, and submit your "Accept" recommendation.

Reviewer #2: All comments have been addressed

Reviewer #3: (No Response)

Reviewer #4: (No Response)

Reviewer #5: All comments have been addressed

2. Does this manuscript meet PLOS Global Public Health’s publication criteria ? Is the manuscript technically sound, and do the data support the conclusions? The manuscript must describe methodologically and ethically rigorous research with conclusions that are appropriately drawn based on the data presented.

Reviewer #2: Yes

Reviewer #3: (No Response)

Reviewer #4: Yes

Reviewer #5: Yes

3. Has the statistical analysis been performed appropriately and rigorously?

Reviewer #2: Yes

Reviewer #3: (No Response)

Reviewer #4: Yes

Reviewer #5: Yes

4. Have the authors made all data underlying the findings in their manuscript fully available (please refer to the Data Availability Statement at the start of the manuscript PDF file)?

Reviewer #2: Yes

Reviewer #3: (No Response)

Reviewer #4: Yes

Reviewer #5: Yes

5. Is the manuscript presented in an intelligible fashion and written in standard English?

Reviewer #2: Yes

Reviewer #3: Yes

Reviewer #4: Yes

Reviewer #5: Yes

6. Review Comments to the Author

Reviewer #2: I appreciate the substantial revisions made by the authors in response to my feedback. The efforts to clarify and expand on key methodological and contextual points have significantly enhanced the manuscript. The added detail in the introduction provides essential clarity, particularly the breakdown of transmission modes, which strengthens the study’s position within the broader infectious disease modeling field. Incorporating recent studies, contextualizes the study’s relevance to contemporary public health issues effectively.

Furthermore, the authors’ explanation of the Bayesian framework and the choice of the beta-binomial model now offers greater accessibility to readers with varying levels of statistical expertise. The expanded Materials and Methods section, detailing the rationale behind prior selections, enhances transparency and methodological robustness. The careful response regarding hierarchical modeling and handling of zero-probability departures demonstrates a strong commitment to methodological rigor, balancing innovation with practical considerations in data analysis.

Overall, I recommend accepting this manuscript for publication, as the revisions made have considerably enhanced its clarity, methodological rigor, and impact. The authors are to be commended for their thorough and thoughtful responses, which I am confident will provide valuable insights for the field.

Reviewer #3: A thorough proofreading might be important to correct minor grammatical errors and typos observed throughout the manuscript

Reviewer #4: Major Strengths:

Novel approach to integrating multiple mobility datasets using a Bayesian framework.

Comprehensive comparison of four different mobility data sources (travel survey, DHS, mobile phone data, Facebook data).

Demonstration of how mobility data source selection impacts measles transmission modeling results.

Rigorous statistical methods and sensitivity analyses.

Important implications for infectious disease modeling and public health planning.

Minor Weaknesses:

Some sections of the methods could be clarified or expanded slightly.

Additional discussion of limitations would strengthen the paper.

A few minor grammatical and formatting issues to address.

Introduction:

Well-written and provides appropriate context and rationale for the study.

Clear objectives are stated.

Methods:

Overall thorough and well-described.

The weighting approach for mobile phone and Facebook data could be explained in slightly more detail in the main text.

Consider adding a brief explanation of why measles was chosen as the case study disease.

Results:

Clearly presented and logically organized.

Figures and tables are informative and well-designed.

Consider adding confidence intervals to some of the key numerical results.

Discussion:

Insightful interpretation of results and their implications.

Could be strengthened by:

Further discussing limitations of the study and datasets used.

Expanding on potential applications of the integrated mobility estimates.

Suggesting future research directions.

Minor Issues:

A few grammatical errors and typos to correct (e.g., "ninemonth" should be "nine-month").

Ensure consistent formatting of headings throughout.

Double-check all in-text citations match the reference list.

Reviewer #5: The study addresses an important issue in infectious disease modeling by evaluating and integrating multiple sources of human mobility data (mobile phone records, Facebook data, travel surveys, and DHS data) to model measles virus transmission in Zambia. The work explores the impact of each data source on estimates of disease transmission and uses a Bayesian framework to integrate these data, ultimately demonstrating that the choice of mobility data can significantly alter predictions.

The manuscript appears well-structured, thorough, and presents a novel approach by comparing multiple mobility data sources and integrating them using a Bayesian framework. The study is timely and relevant, especially given the growing use of mobility data in epidemiology, particularly after the COVID-19 pandemic. This manuscript is suitable for publication, but it may require some revisions before acceptance.

The revisions required can be classified as minor rather than major, primarily focused on:

Some sections, particularly those describing the Bayesian modeling approach and simulation, could benefit from clearer explanations or additional detail for readers who may not be familiar with the technical methods used. Streamlining and reducing redundancy in the introduction and discussion sections would improve readability.

More detail on the parameterization of the metapopulation model and sensitivity analyses would help ensure reproducibility. For example, providing more specifics on how the priors were set for the Bayesian model and how they influence the outcomes would be beneficial. Including a visual representation of the Bayesian framework and the data integration process could enhance the clarity of the methods.

While the manuscript discusses limitations related to sampling biases, especially regarding mobile phone and Facebook data, it could be further strengthened by explicitly addressing how these biases might affect the generalizability of the findings. For instance, the demographic differences in mobile phone ownership (age, gender, socio-economic status) could significantly impact the conclusions drawn.

A more in-depth discussion on the potential biases introduced by using aggregated Facebook data, especially during the COVID-19 pandemic, and how this might influence mobility patterns is warranted. Although the ethical considerations regarding data use are addressed, emphasizing how data privacy concerns were mitigated, particularly with Facebook and mobile phone data, would be important for transparency.

The limitations are well-discussed, particularly regarding the biases inherent in different data sources and the challenges of integrating them. However, a deeper exploration of how these limitations might affect public health policy decisions based on the findings would provide more context for the study's real-world applications.

Conclusion - overall, the manuscript is a solid contribution to the field of infectious disease modeling, with a focus on improving the accuracy of mobility data integration. With minor revisions to improve clarity and expand on certain methodological and ethical considerations, it would be ready for publication.

Overall Assessment - the revisions are sufficiently thorough and address the key concerns raised by the reviewers. Thus, the manuscript is now suitable for publication, with only minor revisions potentially needed for further clarity or editorial adjustments. The limitations have been discussed adequately, making the manuscript well-prepared for publication.

7. PLOS authors have the option to publish the peer review history of their article (what does this mean? ). If published, this will include your full peer review and any attached files.

**Do you want your identity to be public for this peer review?** For information about this choice, including consent withdrawal, please see our Privacy Policy .

Reviewer #2: **Yes: ** Baran Erdik, MD, MHPA

Reviewer #3: **Yes: ** Dr Adamu Zerihun Gelaw

Reviewer #4: **Yes: ** Benjamin Djoudalbaye

Reviewer #5: No

---

## [Editor Report · Decision Letter 2]

2 Jan 2025

PGPH-D-24-01677R2

Comparing and integrating human mobility data sources for measles transmission modeling in Zambia

Dear Natalya,

Thank you for submitting your manuscript to PLOS Global Public Health. After careful consideration, we feel that it has merit but does not fully meet PLOS Global Public Health’s publication criteria as it currently stands. Therefore, we invite you to submit a revised version of the manuscript that addresses the points raised during the review process.

We look forward to receiving your revised manuscript.

Kind regards,

Collins Otieno Asweto, PhD

Academic Editor
---

## [Decision Letter · Decision Letter 3]

20 Mar 2025

PGPH-D-24-01677R3

Comparing and integrating human mobility data sources for measles transmission modeling in Zambia

Dear Natalya,

Thank you for submitting your manuscript to PLOS Global Public Health. After careful consideration, we feel that it has merit but does not fully meet PLOS Global Public Health’s publication criteria as it currently stands. Therefore, we invite you to submit a revised version of the manuscript that addresses the points raised during the review process.

We look forward to receiving your revised manuscript.

Kind regards,

Collins Otieno Asweto, PhD

Academic Editor

Journal Requirements:

Reviewer's Responses to Questions

**Comments to the Author**

1. If the authors have adequately addressed your comments raised in a previous round of review and you feel that this manuscript is now acceptable for publication, you may indicate that here to bypass the “Comments to the Author” section, enter your conflict of interest statement in the “Confidential to Editor” section, and submit your "Accept" recommendation.

Reviewer #1: All comments have been addressed

Reviewer #2: All comments have been addressed

Reviewer #4: All comments have been addressed

Reviewer #6: (No Response)

2. Does this manuscript meet PLOS Global Public Health’s publication criteria ? Is the manuscript technically sound, and do the data support the conclusions? The manuscript must describe methodologically and ethically rigorous research with conclusions that are appropriately drawn based on the data presented.

Reviewer #1: Yes

Reviewer #2: Yes

Reviewer #4: Yes

Reviewer #6: (No Response)

3. Has the statistical analysis been performed appropriately and rigorously?

Reviewer #1: Yes

Reviewer #2: Yes

Reviewer #4: Yes

Reviewer #6: (No Response)

4. Have the authors made all data underlying the findings in their manuscript fully available (please refer to the Data Availability Statement at the start of the manuscript PDF file)?

Reviewer #1: Yes

Reviewer #2: Yes

Reviewer #4: Yes

Reviewer #6: (No Response)

5. Is the manuscript presented in an intelligible fashion and written in standard English?

Reviewer #1: Yes

Reviewer #2: Yes

Reviewer #4: Yes

Reviewer #6: (No Response)

6. Review Comments to the Author

Reviewer #1: The manuscript "Comparing and integrating human mobility data sources for measles transmission modeling in Zambia" has met all the requirements to be considered acceptable for publication. I would recommend the manuscript for PLOS Global Public Health publication.

Reviewer #2: Thank you for adding the legends as well as clarifying the License. Humdata states all data is CC BY 4 unless otherwise noted, which in my opinion should satisfy PLOS requirements.

Reviewer #4: Recommendations for Publication

Clarify Data Integration Challenges: Provide more detailed explanations of the technical challenges encountered during data integration and how they were addressed.

Expand Discussion on Public Health Implications: Further elaborate on how the findings can inform public health policy and disease control strategies beyond Zambia.

Consider Additional Data Sources: Discuss potential future directions for incorporating other mobility data sources, such as GPS loggers or transportation flows, to enhance model accuracy.

Overall, the article contributes significantly to the field of infectious disease modeling by highlighting the importance of integrating diverse data sources to improve the accuracy of mobility estimates and disease transmission dynamics. With minor revisions to address the suggested areas, the article would be well-suited for publication in PLOS Global Public Health.

Reviewer #6: The concept of this manuscript is a bit hard to grasp. If it were explained in simpler terms, it would be much easier for general readers to understand the idea.

7. PLOS authors have the option to publish the peer review history of their article (what does this mean? ). If published, this will include your full peer review and any attached files.

**Do you want your identity to be public for this peer review?** For information about this choice, including consent withdrawal, please see our Privacy Policy .

Reviewer #1: No

Reviewer #2: **Yes: ** Baran Erdik, MD, MHPA

Reviewer #4: **Yes: ** Djoudalbaye Benjamin

Reviewer #6: No

---

## [Decision Letter · Decision Letter 4]

10 Apr 2025

Comparing and integrating human mobility data sources for measles transmission modeling in Zambia

PGPH-D-24-01677R4

Dear Natalya,

We are pleased to inform you that your manuscript 'Comparing and integrating human mobility data sources for measles transmission modeling in Zambia' has been provisionally accepted for publication in PLOS Global Public Health.

Best regards,

Collins Otieno Asweto, PhD

Academic Editor

Reviewer's Responses to Questions

**Comments to the Author**

1. If the authors have adequately addressed your comments raised in a previous round of review and you feel that this manuscript is now acceptable for publication, you may indicate that here to bypass the “Comments to the Author” section, enter your conflict of interest statement in the “Confidential to Editor” section, and submit your "Accept" recommendation.

Reviewer #1: All comments have been addressed

Reviewer #2: All comments have been addressed

2. Does this manuscript meet PLOS Global Public Health’s publication criteria ? Is the manuscript technically sound, and do the data support the conclusions? The manuscript must describe methodologically and ethically rigorous research with conclusions that are appropriately drawn based on the data presented.

Reviewer #1: Yes

Reviewer #2: Yes

3. Has the statistical analysis been performed appropriately and rigorously?

Reviewer #1: Yes

Reviewer #2: Yes

4. Have the authors made all data underlying the findings in their manuscript fully available (please refer to the Data Availability Statement at the start of the manuscript PDF file)?

Reviewer #1: Yes

Reviewer #2: Yes

5. Is the manuscript presented in an intelligible fashion and written in standard English?

Reviewer #1: Yes

Reviewer #2: Yes

6. Review Comments to the Author

Reviewer #1: The manuscript, " Comparing different sources of human mobility data and methods of integration:

modeling measles virus transmission in Zambia" has fulfilled all the recommendations of the reviewers to best of my knowledge and therefore merit acceptance for publication in the PLOS Global Public Health Journal. The manuscript will serve as an avenue of novel scientific knowledge to explore by others in advancing human mobility data. I hereby recommend the manusceipt for publication.

Reviewer #2: Thank you again for your revision. All my comments have been since the second revision.

7. PLOS authors have the option to publish the peer review history of their article (what does this mean? ). If published, this will include your full peer review and any attached files.

**Do you want your identity to be public for this peer review?** For information about this choice, including consent withdrawal, please see our Privacy Policy .

Reviewer #1: **Yes: ** Sikiru Olanrewaju Badaru

Reviewer #2: No
